# Improved Black Widow Spider Optimization Algorithm Integrating Multiple Strategies

**DOI:** 10.3390/e24111640

**Published:** 2022-11-11

**Authors:** Chenxin Wan, Bitao He, Yuancheng Fan, Wei Tan, Tao Qin, Jing Yang

**Affiliations:** 1Electrical Engineering College, Guizhou University, Guiyang 550025, China; 2Power China Guizhou Engineering Co., Ltd., Guiyang 550001, China; 3College of Forestry, Guizhou University, Guiyang 550025, China

**Keywords:** black widow spider optimization algorithm, Gauss chaotic map, sine and cosine strategy, elite opposition-based learning, differential evolutionary algorithm

## Abstract

The black widow spider optimization algorithm (BWOA) had the problems of slow convergence speed and easily to falling into local optimum mode. To address these problems, this paper proposes a multi-strategy black widow spider optimization algorithm (IBWOA). First, Gauss chaotic mapping is introduced to initialize the population to ensure the diversity of the algorithm at the initial stage. Then, the sine cosine strategy is introduced to perturb the individuals during iteration to improve the global search ability of the algorithm. In addition, the elite opposition-based learning strategy is introduced to improve convergence speed of algorithm. Finally, the mutation method of the differential evolution algorithm is integrated to reorganize the individuals with poor fitness values. Through the analysis of the optimization results of 13 benchmark test functions and a part of CEC2017 test functions, the effectiveness and rationality of each improved strategy are verified. Moreover, it shows that the proposed algorithm has significant improvement in solution accuracy, performance and convergence speed compared with other algorithms. Furthermore, the IBWOA algorithm is used to solve six practical constrained engineering problems. The results show that the IBWOA has excellent optimization ability and scalability.

## 1. Introduction

Meta-heuristic algorithms are a class of algorithms that seek to optimize solutions by simulating natural and human intelligence. Swarm intelligence algorithms are a class of meta-heuristic algorithms, which are abstracted by imitating the foraging or other group behaviors of insects, herds, birds and fish. The common swarm intelligence algorithms are: particle swarm optimization (PSO) [1], grey wolf optimization (GWO) [2] butterfly optimization algorithm (BOA) [3], whale optimization algorithm (WOA) [4], cuckoo algorithm (CS) [5] and so on. Swarm intelligence algorithms not only have the advantages of simple implementation, good robustness, easy scalability and self-organization, but can also effectively combine some unique strategies or other algorithms to balance global search and local search capabilities to achieve optimal search.

The balance between local search and global search is the core and key of studying swarm intelligence algorithms. Its core connotation is to ensure the convergence accuracy and speed of the algorithm and to avoid the algorithm falling into local optimum mode at the same time. For this reason, many scholars have made corresponding improvements to the intelligent optimization algorithms they study. For example, Xu et al. [6] and Liu et al. [7] improved their algorithms by using the ergodicity and randomness of Gauss map to initialize the population, avoiding the influence of uncertainty caused by random population. The algorithm has a wider search range and lays a good foundation for global optimization. Kuo et al. [8] used a multi-objective sine cosine algorithm for sequence deep clustering and classification. The performance of the objective function is improved by using the good global development ability of the sine cosine algorithm. Clustering error and classification accuracy achieved superior performance compared with other algorithms. Mookiah et al. [9] proposed an enhanced sine cosine algorithm. Using the sine cosine algorithm forces the local optimal value out to determine the optimal threshold of color image segmentation. Finally, good experimental results were obtained. When Yuan et al. [10] and Zhou et al. [11] improved their algorithms, they all introduced the elite opposition-based learning strategy, which makes full use of individuals with better performance to optimize the next generation. The convergence speed and stability of the algorithm are improved. The above strategies have achieved good results in different algorithms. This also brings enlightenment to the algorithm improvement of this paper. However, the effect of the above strategies applied to the same algorithm had not been tested. This paper considers introducing the above strategy into an algorithm at the same time, and the performance of the improved algorithm is tested.

The black widow spider optimization algorithm (BWOA) was proposed in 2020 by Peña-Delgado et al. [12], inspired by the unique mating behavior of the black widow spider. The algorithm simulated the different behaviors of black widow spiders during courtship. Compared with existing optimization algorithms, the principle and structure of BWOA are relatively simple, and fewer parameters need to be adjusted. However, the algorithm itself still has some shortcomings. For example, for some complex optimization tasks, the traditional BWOA suffers from premature convergence or easily falls into local optimum mode. In addition, the convergence speed of the BWOA is not high enough to obtain high precision solutions for complex problems. Therefore, to address the problems above, this paper improves the original BWOA algorithm and proposes a multi-strategy black widow spider optimization algorithm (IBWOA).

To verify the effectiveness of each improvement strategy and the performance of the proposed algorithm, 13 benchmark functions and a part of CEC2017 were tested. The optimization results are compared and statistically analyzed with other well-known metaheuristic algorithms. Moreover, the IBWOA is used to solve six practical constrained engineering problems, including welded beam design [2], tension spring design [2], three-bar truss design [5], cantilever design [5], I-beam design [5] and tubular column design [5]. In general, the main highlights and contributions of this paper are summarized as follows: (i) a multi-strategy black widow spider optimization algorithm (IBWOA) is proposed, (ii) a proposed approach to optimize the 13 benchmark test functions and a part of CEC2017 test functions is used, which is compared with many typical meta-heuristic algorithms and (iii) the proposed approach to solve six constrained engineering problems is used, which is then compared with many advanced methods.

The rest of this paper is organized as follows: Section 2 presents the mathematical model of the original BWOA. In Section 3, some improved strategies are introduced and integrated into the original algorithm. The IBWOA is proposed and its time complexity is analyzed. Section 4 illustrates the comparative analysis for solving the numerical optimization, and the experimental results are also performed in detail. In Section 5, the IBWOA is used to deal with six practical constrained engineering problems, which compares the IBWOA with various optimization algorithms for optimization testing. Finally, the conclusions and future studies are summarized in Section 6.

## 2. Basic Black Widow Spider Optimization Algorithm

This section introduces the different courtship-mating movement strategies and mathematical models of pheromone rates in black widow spiders.

### 2.1. Movement

The black widow spider moves within the spider web in a linear and spiral fashion. The mathematical model can be formulated as follows:(1)$${\vec{x}}_{i}\left(t+1\right)={\vec{x}}_{*}\left(t\right)-m{\vec{x}}_{r1}\left(t\right)$$
(2)$${\vec{x}}_{i}\left(t+1\right)={\vec{x}}_{*}\left(t\right)-\cos\left(2\pi \beta \right){\vec{x}}_{i}\left(t\right)$$
where ${\vec{x}}_{i}\left(t+1\right)$ is the individual position after the update and ${\vec{x}}_{*}\left(t\right)$ is the current optimal individual position. Random numbers are generated directly or indirectly using the rand function (generates random numbers between 0 and 1). m is a random floating-point number in [0.4, 0.9]. β is a random number in [−1, 1]. $r_{1}$ is a random integer between 1 and the maximum population size. ${\vec{x}}_{r1}\left(t\right)$ is the randomly selected position $r_{1}$, and ${\vec{x}}_{i}\left(t\right)$ is the current individual position.

The way black widow spiders move is determined by random numbers. When the random number generated by the rand function is less than or equal to 0.3, the individual movement mode selects Equation (1), otherwise, the individual movement mode selects Equation (2).

### 2.2. Sex Pheromones

Sex pheromones play a very important role in the courtship process of black widow spiders. Well-nourished female spiders produce more silk than starving females. Male spiders are more responsive to sex pheromones from well-nourished female spiders because they provide a higher level of fertility, so that male spiders primarily avoid the cost of risking mating with potentially hungry female spiders. Therefore, male spiders do not prefer females with low sex pheromones levels. [12] The sex pheromones rate value of the black widow spider is defined as:(3)$$pheromone\left(i\right)=\frac{fitness_{\max}-fitness\left(i\right)}{fitness_{\max}-fitness_{\min}}$$
where $fitness_{\max}$ and $fitness_{\min}$ represent the worst and best fitness values in the current population,  fitness(i) is the fitness value of the individual i. The sex pheromones vector contains normalized fitness in [0,1]. For individuals with sex pheromones rates less than or equal to 0.3, the position update method can be formulated as follows:(4)$${\vec{x}}_{i}\left(t\right)={\vec{x}}_{*}\left(t\right)+\frac{1}{2}\left[{\vec{x}}_{r1}\left(t\right)-{\left(-1\right)}^{\sigma }{\vec{x}}_{r2}\left(t\right)\right]$$
where ${\vec{x}}_{i}\left(t\right)$ is the position of female black widow spiders with low sex pheromones levels. $r_{1}$ and $r_{2}$ are random integers from 1 to the maximum population size, and $r_{1}\neq r_{2}$. σ is a random binary number in {0, 1}.

## 3. Improvements to the Black Widow Spider Optimization Algorithm

In this section, some improved strategies are introduced and integrated into the original algorithm. The IBWOA is proposed and its time complexity is analyzed.

### 3.1. Gauss Chaos Mapping to Initialize the Population

Shan Liang et al. [13] found and proposed that when the initial sequence of positions is uniformly distributed in the search space, it can effectively improve the algorithm optimization performance. The original black widow spider algorithm directly uses the rand function to initialize the population. This generates populations with high randomness, but which are not necessarily uniformly distributed throughout the solution space. This leads to a slow population search and insufficient algorithmic diversity. To address this problem, Gauss chaotic mapping is introduced to initialize the population and improve the diversity of the algorithm. It enables the algorithm to quickly discover the location of high-quality solutions, thus speeding up the convergence speed of the algorithm and improving the convergence accuracy of the algorithm.

Gauss mapping is a classical mapping of one-dimensional mappings, and it is defined as:(5)$$z_{n+1}=\{\begin{array}{c} \begin{array}{cc} 0, & z_{n}=0 \end{array}\\ \begin{array}{cc} \frac{1}{z_{n}\mathrm{mod}\left(1\right)}, & z_{n}\neq 0 \end{array} \end{array}$$
(6)$$\frac{1}{z_{n}\mathrm{mod}(1)}=\frac{1}{z_{n}}-\left[\frac{1}{z_{n}}\right]$$
where mod is the residual function and [ ] denotes rounding and $z_{1},z_{2},\ldots ,z_{n}$ is the chaotic sequence generated by the Gauss mapping. The BWOA after introducing Gauss mapping to initialize the population is denoted as GBWOA.

The comparison between (**a**) and (**b**) in Figure 1 shows that Gauss chaotic mapping produces a more uniform population distribution and a higher quality population.

### 3.2. Sine and Cosine Strategy

The sine cosine algorithm (SCA) is a novel nature-like optimization algorithm proposed by Seyedali Mirjalili in 2016 [14]. The algorithm creates multiple random candidate solutions. The mathematical properties of the sine and cosine functions are used to adaptively change the amplitudes of the sine and cosine functions. In turn, the algorithm balances global exploration and local exploitation capabilities in the search process and eventually finds the global optimal solution. Its update can be formulated as follows:(7)$${\vec{x}}_{i}\left(t+1\right)=\vec{x}\left(t\right)+l_{1}\cdot \sin l_{2}\cdot \left|l_{3}{\vec{\cdot x}}_{*}\left(t\right)-\vec{x}\left(t\right)\right|$$
(8)$${\vec{x}}_{i}\left(t+1\right)=\vec{x}\left(t\right)+l_{1}\cdot \cos l_{2}\cdot \left|l_{3}{\vec{\cdot x}}_{*}\left(t\right)-\vec{x}\left(t\right)\right|$$
where ${\vec{x}}_{i}\left(t+1\right)$ is the individual position after updating. ${\vec{x}}_{*}\left(t\right)$ is the current optimal individual position. $l_{2}$ is a random number in [0, 2π], and $l_{3}$ is a random number in [0,2]. ${\vec{x}}_{i}\left(t\right)$ is the current individual position.

$l_{4}$ is a random number in [0, 1]. When $l_{4}<0.5$, the position update is performed using Equation (8), otherwise, the position is updated using Equation (9). $l_{1}$ is determined by the following equation:(9)$$l_{1}=a\cdot \left(1-\frac{t}{T}\right)$$
where a is a constant generally taking the value of 2. t is the number of current iterations, and T is the maximum number of iterations.

The random number is generated by the rand function to be less than or equal to the mutation probability p to perform the mutation. The mutation probability p can be formulated as follows:(10)$$p=\exp{\left(1-\frac{t}{T}\right)}^{-20}+0.35$$

Suppose that the maximum number of iterations T is 500, and so the variation trend of the mutation probability is shown as follows:

It can be seen from Figure 2 that the introduction of mutation probability p controls the weight of the algorithm to perform mutation. The probability of performing mutation operation is higher in the middle and early stages of the algorithm iteration. The probability of performing mutation in the later part of the algorithm iteration is smaller or even 0. The sine cosine algorithm is introduced as a variance perturbation strategy to the original BWOA, and is denoted as SBWOA.

### 3.3. Elite Opposition-Based Learning

Opposition-based learning (OBL) is an intelligent technique proposed by Tizhoosh [15]. Its main idea is to evaluate both the current solution and its opposite solution and use them in a meritocratic way in order to enhance the search range and capability of the algorithm. Later, Wang et al. [16] further proposed the concept of general opposition-based learning. Wang S.W et al. [17] proposed an elite opposition-based learning strategy (EO) based on the general opposition-based learning strategy. The experimental results show that the elite opposition-based learning strategy has better performance than the general opposition-based learning strategy.

The elite opposition-based learning strategy merges the opposite population with the current population and selects the best individuals into the next generation population. It enhances the diversity of the population and reduces the probability of the algorithm falling into local optimum. At the same time, it fully absorbs the useful search information of the elite individuals in the current population. Therefore, it can accelerate the convergence speed of the algorithm.

**Definition** **1.***Suppose that*$x_{i}\left(k\right)$*and*$x_{i}^{*}\left(k\right)$*are the current solutions and the opposition solutions of the generation*k. $x_{i,j}\left(k\right)$*and*$x_{i,j}^{*}\left(k\right)$*are values on dimension*j*of*$x_{i}\left(k\right)$*and*$x_{i}^{*}\left(k\right)$*, respectively.*e (2≤e≤N)*elite individuals are denoted as:*$\{e_{1}\left(k\right),e_{2}\left(k\right),\cdots e_{e}\left(k\right)\}\subseteq \{x_{1}\left(k\right),x_{2}\left(k\right),\cdots x_{N}\left(k\right)\}$*, and can then*$x_{i,j}^{*}\left(k\right)$*be defined as:*(11)$$x_{i,j}^{*}\left(k\right)=\lambda \left(a_{j}\left(k\right)+b_{j}\left(k\right)\right)-x_{i,j}\left(k\right)$$*where*$a_{j}\left(k\right)=\min\left(e_{1,j}\left(k\right),\cdots ,e_{e,j}\left(k\right)\right),b_{j}\left(k\right)=\max\left(e_{1,j}\left(k\right),\cdots ,e_{e,j}\left(k\right)\right)$. λ*is a random number in*(0,1)*. Set the out-of-bounds treatment as follows: if*$x_{i,j}^{*}\left(k\right)>b_{j}\left(k\right)$*, then take*$x_{i,j}^{*}\left(k\right)=b_{j}\left(k\right)$*; if*$x_{i,j}^{*}\left(k\right)<a_{j}\left(k\right)$*, then take*$x_{i,j}^{*}\left(k\right)=a_{j}\left(k\right)$.

Research shows that the elite opposition-based learning strategy exhibits the best performance when e=0.1×N [17]. The elite opposition-based learning strategy is introduced into the original BWOA notated as EBWOA and is executed at the end of each iteration.

### 3.4. Differential Evolution Algorithm

The differential Evolution (DE) algorithm was proposed in 1997 by Rainer Storn and Kenneth Price [18] on the basis of the genetic algorithm (GA). The variation can be formulated as follows:(12)$${\vec{x}}_{i}\left(t\right)={\vec{x}}_{r1}\left(t\right)+F\cdot \left({\vec{x}}_{r2}\left(t\right)-{\vec{x}}_{r3}\left(t\right)\right)$$
where ${\vec{x}}_{r1}\left(t\right),{\vec{x}}_{r2}\left(t\right),{\vec{x}}_{r3}\left(t\right)$ are three individual positions randomly selected from the current population and are different from each other. F is the scaling factor. too small an F may cause the algorithm to fall into a local optimum, and too large an algorithm does not converge easily. Therefore, F is usually taken as a random number between [0.4,1].

Combine the principle of BWOA, where the position update is guided by the current optimal individual. Replace the random individual position ${\vec{x}}_{r1}\left(t\right)$ in Equation (11) with the current optimal individual position ${\vec{x}}_{*}\left(t\right)$. For individuals with sex pheromones rate values less than or equal to 0.3 in BWOA, the new individual position update can be formulated as follows:(13)$${\vec{x}}_{i}\left(t\right)={\vec{x}}_{*}\left(t\right)+F\cdot \left({\vec{x}}_{r1}\left(t\right)-{\vec{x}}_{r2}\left(t\right)\right)$$

Comparing Equation (4), it can be seen that after combining the mutation principle of the differential evolution algorithm, the individual position updating method removes the random binary number σ and constitutes a strict differential vector ${\vec{x}}_{r1}\left(t\right)-{\vec{x}}_{r2}\left(t\right)$. The introduction of the scaling factor F to replace the fixed constant 0.5 in Equation (4) makes the position update method more random and diverse. This operation is more conducive to the recombination of individuals with poor fitness values and to the full utilization of the population resources. Equation (13) was introduced to replace Equation (4) with the original BWOA, noted as DBWOA.

### 3.5. Time Complexity Analysis

The time complexity of the BWOA is O(N×d×Max_iter), where N is the population size, d is the dimensionality and Max_iter is the maximum number of iterations.

The DBWOA is a modification of the original algorithm in a variant way, so the time complexity is unchanged.

The time complexity of introducing Gauss chaos mapping sequence to initialize the population is O(N×d). The time complexity of the GBWOA for introducing a Gauss chaotic mapping sequence to initialize the population can be formulated as follows:(14)O(N×d×Max_iter+N×d)=O(N×d×Max_iter)

Introduce the sine cosine strategy. The mutation perturbation update position cost is O(Max_iter)O(N×d). The time complexity of the SBWOA with the introduction of the sine cosine strategy can be formulated as follows:(15)O(N×d×Max_iter)+O(Max_iter)O(N×d)=O(N×d×Max_iter)

Introduce the elite opposition-based learning strategy. The cost of calculating the fitness value of each individual is O(N×d×f), where f represents the cost of the objective function. The cost of obtaining the elite opposition-based solution is O(N×d). The cost of quick sort is $O\left(N^{2}\right)$. The time complexity of the EBWOA with the introduction of the sine and cosine strategy can be formulated as follows:(16)$$O\left(N\times d\times Max\_iter\right)+O\left(Max\_iter\right)O\left(N^{2}+N\times d\times \left(f+1\right)\right)$$

The IBWOA introduces and integrates many of these improvement strategies mentioned above. Initialize the population using Gauss mapping, which integrates the mutation approach of the differential evolution algorithm. Randomly select one of the sine cosine and elite opposition-based learning strategies to execute. Ensure the convergence speed of the algorithm, while performing as many mutation perturbations as possible. Help the algorithm to better jump out of the local optimum mode and improve the accuracy of the test results. The time complexity of the IBWOA can be formulated as follows:(17)$$O\left(N\times d\times Max\_iter\right)+O\left(c\right)O\left(N^{2}+N\times d\times \left(f+1\right)\right)$$
where c (c<Max_iter) represents the number of elite opposition-based learning executions. The pseudo-code of the proposed algorithm is shown in Algorithm 1.
**Algorithm 1:** The pseudo-code of IBWOAInitializing populations using Gauss chaos mappingCalculate the fitness value of each spider Record the current worst fitness value, the best fitness value and its location information while $t<T_{\max}$
 initialize random parameters $m,\beta ,p,l_{1},l_{2},l_{3},l_{4}$
for i=1:N
  if random ≤ 0.3  the spider moves and update its location information using Equation (1)  or else  the spider moves and update its location information using Equation (2)  end if  calculating the pheromone value of the spider using Equation (3)  update the spider with low pheromone values using Equation (13)   if random ≤  p
   h = 0   if ≤ 0.5   update the spider location information using Equation (7)   or else   update the spider location information using Equation (8)   end if   or else   h = 1   end if  calculate the fitness value of the spider  if the fitness value of the spider ≤ the best fitness value,  update the best fitness value and its location information  end if end for if h == 1 Obtain opposition solutions using Equation (11) Retaining spiders with higher fitness values end ift=t+1end whileOutput the best fitness value

## 4. Results of Experiments

In this section, the performance of the IBWOA is substantiated extensively. To verify the effectiveness of each improvement strategy and the performance of the proposed algorithm, 13 benchmark functions and a part of CEC2017 were tested. Moreover, the optimization results are compared and statistically analyzed with other well-known metaheuristic algorithms.

The simulation environment is: Intel Core i5-8400 CPU with 2.80 GHz, Windows 11 64-bit operating system and simulation software Matlab2017 (b).

### 4.1. Introduction of Benchmarking Functions

In order to test the performance of IBWOA, 13 benchmark test functions used in the literature [19] were selected for the optimization test, where $f_{1}$−$f_{4}$ are unimodal functions, $f_{5}$−$f_{10}$ are multimodal functions and $f_{11}$−$f_{13}$ are fixed-dimensional functions. The information related to the benchmark test function is shown in Table 1.

### 4.2. Comparison of the Optimization Results of Each Improvement Strategy

In order to test the optimization effect of different improvement strategies and verify the rationality and effectiveness of each improvement strategy, four typical benchmark test functions, $f_{1}$, $f_{3}$, $f_{5}$ and $f_{9}$, are selected and run 30 times independently in different dimensions for the BWOA, and GBWOA, SBWOA, EBWOA and DBWOA. The mean and standard deviation of the optimized test results were recorded. The population number of each algorithm was set to 30, and the max iteration was set to 500. The test results are given in Table 2, and the search curve of each improvement strategy in 30 dimensions is given in Figure 3, where the x-axis represents the number of iterations of the algorithms and the y-axis represents the logarithmic form of the fitness values.

After each improvement strategy is applied to the algorithm, the performance of the search in different dimensions remains basically the same.

After introducing Gauss mapping to initialize the population, the improvement effect of the GBWOA on unimodal functions $f_{1}$ and $f_{3}$ are not obvious. However, for multimodal functions $f_{5}$ and $f_{9}$, the search accuracy is significantly improved. It shows that Gauss mapping improves the diversity at the beginning of the population and can better jump out of the local optimum mode. It is helpful for improving the search accuracy of the algorithm.

In the optimization test for unimodal functions $f_{1}$ and $f_{3}$, after the introduction of the elite opposition-based learning strategy, the EBWOA is able to converge quickly to the theoretical optimal value 0 in all dimensions. Compared with the test results of the original BWOA, the improvement of the optimization effect is very obvious. The EBWOA also improves the search accuracy for the multimodal functions $f_{5}$ and $f_{9}$, and achieves better results than the original algorithm. However, the result is still quite far from the theoretical optimal value. It shows that the introduction of the elite opposition-based learning strategy can improve the convergence speed and the search accuracy of the algorithm.

In the optimization test for multimodal functions $f_{5}$ and $f_{9}$, the SBWOA converges to near the theoretical optimal value in all dimensions after the introduction of the sine cosine strategy. Compared with the test results of the original BWOA, the improvement of the optimization effect is very obvious. In the optimization test for unimodal functions $f_{1}$ and $f_{3}$, the search accuracy is reduced compared with the original algorithm instead. It shows that the introduction of the sine cosine perturbation can better jump out of the local optimum mode and improve the algorithm’s search accuracy when dealing with the multimodal problem, but it also slows down the convergence speed of the algorithm.

After improving the original BWOA by integrating the mutation of the differential evolution algorithm, the accuracy of the DBWOA for the multimodal function $f_{5}$ in the optimization test is improved compared with the original algorithm, but the improvement for other test functions is not obvious. It is shown that after integrating the mutation of differential evolution to restructure the individuals with poor fitness values, the algorithm has improved the accuracy of the search in dealing with complex multimodal problems.

In summary, the reasonableness and effectiveness of each improvement strategy are verified.

### 4.3. Analysis of Success Rate and Average Running Time of the Algorithm

In order to verify the speed and success rate of IBWOA in handling optimization problems, the BWOA, GBWOA, SBWOA, EBWOA and IBWOA were selected to optimize the benchmark test functions $f_{1}$–$f_{13}$. The success rate and the average running time of per execution of the algorithm are recorded. The population number of each algorithm was set to 30, and the max iteration was set to 500. Each algorithm was run 30 times independently. The success rate of algorithms defined according to the literature [20] can be formulated as:

Assuming that the fitness error is F(t), the mathematical model of F(t) can be formulated as:(18)$$F\left(u\right)=X\left(u\right)-X^{*}$$
where u is the number of times of the algorithm runs. X(u) is the actual optimization result of the algorithm running for the time u. $X^{*}$ is the theoretical optimal value.

The variable δ(u) is defined and its mathematical model can be formulated as:(19)$$\delta \left(u\right)=\{\begin{array}{c} 1,\mathrm{if}\;\left|F\left(u\right)\right|<\varepsilon \\ 0,\mathrm{if}\;\left|F\left(u\right)\right|\geq \varepsilon \end{array}$$
where ε is the fitness error accuracy. The specific value of ε is shown in Table 1.

The mathematical model of $P_{c}$, the success rate of algorithm can be formulated as follows:(20)$$P_{c}=\frac{1}{30}\overset{30}{\underset{u=1}{\sum }}\delta \left(u\right)$$

Defining the variable φ(u) as the running time of the algorithm for the time u, the average running time of each execution of the algorithm Y(Unit is second) can be formulated as follows:(21)$$Y=\frac{1}{30}\sum_{u=1}^{30}\varphi \left(u\right)$$

As shown in Table 3, when testing 13 benchmark functions, the BWOA, GBWOA, DBWOA and SBWOA have a relatively short and almost same average running time per execution. The EBWOA with the introduction of the elite opposition-based learning strategy has the longest average running time per execution. The average running time per execution of the IBWOA after integrating various strategies is increased based on the original algorithm, but is lower than that of EBWOA.

Each algorithm has a relatively short average running time per execution when the algorithms optimize unimodal functions, except $f_{3}$ and fixed-dimension multimodal functions. In the optimization of the unimodal function $f_{3}$, the algorithms have the long average running times per execution. This is related to the solution complexity of the fitness of the objective function itself.

The difference in success rate is mainly reflected in the algorithms optimized functions $f_{4},f_{5},f_{9}$ and $f_{10}$. The GBWOA improves the optimization accuracy of the algorithms, but the improvement is limited. The DBWOA mainly improves the optimization accuracy of the algorithms for multimodal functions, but again the improvement is limited. EBWOA mainly improves the convergence speed of the original algorithm, but the improvement in success rate is not obvious. The SBWOA adds mutation perturbation to help the algorithm jump out of the local optimum, so the success rate of the algorithm on the multimodal function is significantly improved. The IBWOA integrates the advantages of each improvement strategy. Its success rate reaches 100 % and the stability is the best when optimizing 13 benchmark test functions.

### 4.4. Performance Comparison of IBWOA with Other Algorithms

In order to test the optimization performance of the IBWOA for the benchmark test functions, the BWOA, PSO [21], GWO [2], WOA [4], CS [5], BOA [3] and IBWOA, were selected. The 13 benchmark test functions were also selected. The population number of each algorithm was set to 30, and the max iteration was set to 500. The dimension was 30. The optimization test was run 30 times independently, and the mean and standard deviation were recorded. The main parameters of each algorithm are set in Table 4, and the results of the test are shown in Table 5. Figure 4 shows the convergence curves of the seven algorithms used in the experiment on 13 benchmark functions.

For the unimodal test functions $f_{1}–f_{3}$ and the multimodal test functions $f_{6}$ and $f_{8}$, the IBWOA converges to 0, the theoretical optimum value. For the complex multimodal test functions $f_{5}$, the test results of the IBWOA are close to −12,569.48, the theoretical optimum value. The test results of the IBWOA for the functions $f_{4}$, $f_{9}$ and $f_{10}$ also satisfy the allowable absolute error accuracy and are better than the various algorithms compared. For the fixed-dimensional test functions $f_{11}–f_{13}$, the IBWOA also converges to the near theoretical optimal value. In addition, the IBWOA find the global optimum with the smallest standard deviation, reflecting the good robustness of the algorithm.

In summary, the IBWOA has the best results in terms of convergence speed, search accuracy and robustness compared to the other listed algorithms.

### 4.5. Wilcoxon Rank Sum Detection

If only analyzing and comparing the mean and standard deviation of the respective algorithms themselves, such data analysis lacks integrity and scientific validity. In order to further examine the robustness and stability of IBWOA, a statistical analysis method was used: Wilcoxon rank sum detection, which detects complex data and analyzes the performance difference between the IBWOA and other algorithms from a statistical point of view. The PSO, CS, BOA, GWO, WOA and BWOA were selected to optimize 13 benchmark test functions, and the results of each algorithm running independently for 30 times were recorded. Wilcoxon rank sum detection was performed with these data against the results of the IBWOA runs, and P values were calculated. It is set that when P<5%, it can be considered as a strong validation to reject the null hypothesis.

The test results are shown in Table 6, where NaN indicates that there is no data to compare with the algorithm. The +, = and − indicate that the IBWOA outperforms, equals and underperforms against the compared algorithms, respectively.

As shown in Table 6, the results of the Wilcoxon rank sum detection for the IBWOA show that the values for P are overwhelmingly less than 5%. It shows that the optimization advantage of IBWOA for the benchmark function is obvious from the statistical point of view. The robustness of IBWOA is verified.

### 4.6. The Performance of the IBWOA on the CEC2017

Most of the CEC2017 test functions [22] are a combination of the weights of multiple benchmark functions. Such a combination of the weights makes the characteristics of the CEC2017 test functions more complex. These test functions with complex characteristics to test the optimization performance of the IBWOA can be used to further verify the optimization capability and applicability of the IBWOA in the face of complex functions. A part of CEC2017 single-objective optimization functions were selected for the optimization test, which included unimodal (UN), multimodal (MF), hybrid (HF) and composition (CF) functions. The specific information related to the function is given in Table 7.

PRO [23], WOA [23], GWO [23] and BWOA were selected for testing and comparison. The population number of each algorithm was set to 50, and the max iteration was set to 1000. The dimension was 10. Each function runs 30 times independently and the mean and standard deviation are recorded. The results of the optimization test and are given in Table 8.

As shown in Table 8, the IBWOA ranked first in all the results of the optimization test results for the CEC2017 functions, except for CEC05 and CEC06. The results of the optimization test for the unimodal function CEC03 are all far from the theoretical optimum, whereas the test results of the IBWOA are the best compared to the compared algorithms. The results of the optimization test for hybrid and composite functions show that the IBWOA performs more consistently and with higher accuracy. It shows that the performance of the IBWOA is mainly related to the optimization process of hybrid and composition functions, and the IBWOA has great potential in dealing with complex combinatorial problems.

## 5. Practical Constrained Engineering Problems

The penalty function in [24] is selected as the nonlinear constraint condition. In this section, the IBWOA is used to deal with six practical constrained engineering problems, including welded beam design [2], tension spring design [2], three-bar truss design [5], cantilever design [5], I-beam design [5] and tubular column design [5]. The dimensions and constraints of the six constrained engineering problems are given in Table 9. Compare IBWOA with various optimization algorithms for optimization testing. The population number of each algorithm was set to 50, and the max iteration was set to 1000. Each problem runs 30 times independently and the optimal values are recorded.

### 5.1. Welded Beam Design

The welded beam is designed with four main constraints and other lateral constraints. The constraints include shear stress τ, beam bending stress σ, buckling load $P_{c}$, beam deflection δ and other internal parameter constraints.

Its mathematical model of the welded beam design [2] can be formulated as:

Minimize: $f\left(x_{1},x_{1},x_{1},x_{1}\right)=f\left(h,l,t,b\right)=1.10471x_{1}^{2}x_{2}+0.04811x_{3}x_{4}\left(14+x_{2}\right)$

Subject to:$$\begin{array}{c} g_{1}\left(X\right)=\sqrt{{\left(\tau^{\prime }\right)}^{2}+2\tau^{\prime }\tau^{\prime \prime }\frac{x_{2}}{2R}+{\left(\tau^{\prime \prime }\right)}^{2}}-\tau_{\max}\leq 0,\\ g_{2}\left(X\right)=\frac{6PL}{x_{3}^{2}x_{4}}-\sigma_{\max}\leq 0,\\ g_{3}\left(X\right)=x_{1}-x_{4}\leq 0,\\ g_{4}\left(X\right)=0.10471x_{1}^{2}+0.04811x_{3}x_{4}\left(14+x_{2}\right)-5\leq 0,\\ g_{5}\left(X\right)=0.125-x_{1}\leq 0,\\ g_{6}\left(X\right)=\frac{4PL^{3}}{Ex_{3}^{2}x_{4}}-\delta_{\max}\leq 0,\\ g_{7}\left(X\right)=P-\frac{4.013Ex_{3}^{2}x_{4}}{6L^{2}}\left(1-\frac{x_{3}}{2L}\sqrt{\frac{E}{4G}}\right)\leq 0, \end{array}$$
where $x_{1},x_{2},x_{3}$ and $x_{4}$ denote the four basic properties of the welded beam: the weld width h and the width d, length l and thickness b of the beam, respectively. Their range of variation is: $0.1\leq x_{1}\leq 2,\;0.1\leq x_{2}\leq 10,\;0.1\leq x_{3}\leq 10,\;0.1\leq x_{4}\leq 2.$
$$\begin{array}{c} \tau^{\prime }=\frac{P}{\sqrt{2}x_{1}x_{2}},\tau^{\prime \prime }=\frac{MR}{J},M=P\left(L+\frac{x_{2}}{2}\right),P=6000lb,\\ J=2\sqrt{2}x_{1}x_{2}\left[\frac{x_{2}^{2}}{4}+{\left(\frac{x_{1}+x_{3}}{2}\right)}^{2}\right],R=\sqrt{\frac{x_{2}^{2}}{4}+{\left(\frac{x_{1}+x_{3}}{2}\right)}^{2}},\\ L=14\mathrm{in},E=30\times 10^{6}\mathrm{psi},G=12\times 10^{6}\mathrm{psi},\\ \tau_{\max}=13,600\mathrm{psi},\;\sigma_{\max}=30,000\mathrm{psi},\;\delta_{\max}=0.25\mathrm{in}. \end{array}$$

The best solutions for the BWOA, IBWOA, RO [24], CPSO [24], GWO [2], WOA [4], SSA [25]and HFBOA [26] regarding the design of welded beams are given in Table 10. The optimal value of IBWOA is 1.706809, which means that the total cost of the welded beam design is minimized when $x_{1},x_{2},x_{3}$ and $x_{4}$ are set to 0.204300, 3.273201, 9.104938 and 0.205632, respectively. As can be seen from Table 10, the IBWOA obtained the best result in the optimization test among all the compared algorithms.

### 5.2. Tension/Compression Spring Design

Tension/compression spring design [2]. Its objective is to minimize its mass while satisfying certain constraints.

Its mathematical model can be formulated as follows:

Minimize: $f\left(x_{1},x_{2},x_{3}\right)=f\left(d,D,N\right)=\left(x_{3}+2\right)x_{1}^{2}x_{2}$

Subject to:$$\begin{array}{c} g_{1}\left(x\right)=1-\frac{x_{2}^{3}x_{3}}{71785x_{1}^{4}}\leq 0,\\ g_{2}\left(x\right)=\frac{4x_{2}^{2}-x_{1}x_{2}}{12566\left(x_{1}x_{2}^{3}+x_{2}^{4}\right)}+\frac{1}{5108x_{2}^{2}}\leq 0,\\ g_{3}\left(x\right)=1-\frac{140.45x_{2}}{x_{1}^{2}x_{3}}\leq 0,\\ g_{4}\left(x\right)=\frac{x_{1}+x_{2}}{1.5}\leq 0. \end{array}$$
where $x_{1}$, $x_{2}$, and $x_{3}$ represent the spring coil diameter d, spring coil diameter D and the number of windings P, respectively. Their range of variation: $0.25\leq x_{1}\leq 1.3,\;0.05\leq x_{2}\leq 2.0,\;2\leq x_{3}\leq 15.$

The optimal solutions obtained for the BWOA, IBWOA, PSO [21], GWO [2], WOA [4], GSA [27]and HFBOA [26] regarding the design of tension/compression springs are given in Table 11. The optimal value of the IBWOA is 0.012666, which means that the total cost of the tension/compression spring is minimized when $x_{1}$, $x_{2}$ and $x_{3}$ are set as 0.051889, 0.361544 and 11.011088, respectively. As can be seen from Table 11, the results of the IBWOA are better than previous studies, except for the GWO algorithm and HFBOA.

### 5.3. Three-Bar Truss Design

The three-bar truss design [5] problem minimizes the volume while satisfying the stress constraints on each side of the truss member.

Its mathematical model can be formulated as follows:

Minimize: $f\left(x_{1},x_{2}\right)=f\left(A_{1},A_{2}\right)=\left(2\sqrt{2A_{1}}+A_{2}\right)\cdot l$

Subject to:$$\begin{array}{c} g_{1}=\frac{\sqrt{2}x_{1}+x_{2}}{\sqrt{2}x_{1}^{2}+2x_{1}x_{2}}P-\sigma \leq 0\\ g_{1}=\frac{x_{2}}{\sqrt{2}x_{1}^{2}+2x_{1}x_{2}}P-\sigma \leq 0\\ g_{1}=\frac{1}{x_{1}+\sqrt{2}x_{2}}P-\sigma \leq 0 \end{array}$$
where l is the length of the rod truss, and $x_{1}$ and $x_{2}$ denote the cross-sectional area of the long rod truss and short rod truss, respectively. Their range of variation: $0\leq x_{1},x_{2}\leq 1$.
$$l=100\;\mathrm{cm},P=2\;{\mathrm{kN}/\mathrm{cm}}^{2},\sigma =2\;{\mathrm{kN}/\mathrm{cm}}^{2}.$$

The optimal solutions obtained of IBWOA, BWOA, CS [5], SSA [25], HHO [28], MBA [29] and HFBOA [26] regarding the design of the three-bar truss are given in Table 12. The optimal value of the IBWOA is 263.46343425, which means that the total cost of three-bar truss design is minimized when $A_{1}$ and $A_{2}$ are set as 0.786027200 and 0.407114772, respectively. As can be seen from Table 12, the IBWOA obtained the best result for the optimization test among all the algorithms compared.

### 5.4. Cantilever Beam Design

The variables of the cantilever beam design [5] are the height or width of the different beam elements. Their thicknesses are kept fixed in the problem.

The mathematical model can be formulated as follows:

Minimize: $f\left(x_{1},x_{2},x_{3},x_{4},x_{5}\right)=0.0624\left(x_{1}+x_{2}+x_{3}+x_{4}+x_{5}\right)$

Subject to:$$g_{1}=\frac{61}{x_{1}^{3}}+\frac{37}{x_{2}^{3}}+\frac{19}{x_{3}^{3}}+\frac{7}{x_{4}^{3}}+\frac{1}{x_{5}^{3}}-1\leq 0$$
where $x_{1},x_{2},x_{3},x_{4}$ and $x_{5}$ denote the height or width of different beam elements, respectively. Their range of variation: $0.01\leq x_{i}\leq 100,i=1,2,3,4,5$.

The optimal solutions obtained for the BWOA, IBWOA, CS [5], SSA [25], SOS [30], MMA [31] and HFBOA [26] regarding the cantilever beam design are given in Table 13. The optimal value of the IBWOA is 1.307284, i.e., when $x_{1},x_{2},x_{3},x_{4}$ and $x_{5}$ are set as 6.044796, 4.805171, 4.431811, 3.471760 and 2.196531, and the total cost of the cantilever beam is minimized. As can be seen from Table 13, the IBWOA obtained the best result for the optimization test among all the algorithms compared.

### 5.5. I-Beam Design

I-beam design [5], minimal vertical deflection by optimizing length b, height h and both thicknesses $t_{w}$,$\;t_{f}$.

Its mathematical model can be formulated as follows:

Minimize: $f\left(x_{1},x_{2},x_{3},x_{4}\right)=f\left(b,h,t_{w},t_{f}\right)=\frac{5000}{\frac{x_{3}{\left(x_{2}-2x_{4}\right)}^{3}}{12}+\frac{x_{1}x_{4}^{3}}{6}+2x_{1}x_{4}{\left(\frac{x_{2}-x_{4}}{2}\right)}^{2}}$

Subject to:$$g_{1}=2x_{1}x_{3}+x_{3}\left(x_{2}-2x_{4}\right)-300\leq 0$$
where $x_{1},x_{2},x_{3}$ and $x_{4}$ vary in range: $10\leq x_{1}\leq 50,10\leq x_{2}\leq 80,0.9\leq x_{3}\leq 5,0.9\leq x_{4}\leq 5.$

The optimal solutions obtained for the BWOA, IBWOA and CS [5], SOS [30] and CSA [32] regarding the design of I-beams are given in Table 14. The optimal value of the IBWOA is 0.0066260616, when $x_{1},x_{2},x_{3}$ and $x_{4}$ are set as 49.9996, 79.99996414, 1.7644811413, and 4.9999979901, and the total cost of I-beam is minimized. As can be seen from Table 14, the IBWOA obtained the best result for the optimization test among all the algorithms compared.

### 5.6. Tubular Column Design

The goal of the tubular column design [5] is using a minimal cost to obtain a homogeneous column.

Its mathematical model can be formulated as follows:

Minimize: $f\left(x_{1},x_{2}\right)=f\left(d,t\right)=9.8x_{1}x_{2}+2x_{1}$

Subject to:$$\begin{array}{c} g_{1}=\frac{P}{\pi x_{1}x_{2}\sigma_{y}}-1\leq 0\\ g_{2}=\frac{8PL^{2}}{\pi^{3}Ex_{1}x_{2}\left(x_{1}^{2}+x_{2}^{2}\right)}-1\leq 0\\ g_{3}=\frac{2.0}{x_{1}}-1\leq 0\\ g_{4}=\frac{x_{1}}{14}-1\leq 0\\ g_{5}=\frac{0.2}{x_{2}}-1\leq 0\\ g_{6}=\frac{x_{2}}{0.8}-1\leq 0 \end{array}$$
where $x_{1},x_{2}$ denote the average diameter of the column respectively. P is the compressive load, $\sigma_{y}$ is the yield stress, E is the modulus of elasticity, ρ is the density and *L* is the length of the designed column.

Their range of variation: $2\leq x_{1}\leq 14,0.2\leq x_{2}\leq 0.8,P=2500\;\mathrm{kgf},\sigma_{y}=500{\;\mathrm{kgf}/\mathrm{cm}}^{2},E=0.85\times 10^{6}{\;\mathrm{kgf}/\mathrm{cm}}^{2},L=250\;\mathrm{cm},\rho =0.0025{\;\mathrm{kgf}/\mathrm{cm}}^{3}$.

The optimal solutions obtained for the BWOA, IBWOA, CS [5], CSA [32], KH [33] and HFBOA [26] regarding the design of tubular columns are given in Table 15. The optimal value of IBWOA is 26.49633224, which means that the total cost of the tubular column design is the lowest when $x_{1}$ and $x_{2}$ are set as 5.4521171299 and 0.291734575, respectively. As can be seen from Table 15, IBWOA obtained the best result for the optimization test among all the algorithms compared.

## 6. Conclusions

This paper first verifies the reasonableness and effectiveness of each improvement strategy through experiments. Then, the experimental results on success rate show the success rate of the proposed algorithm reaches 100% and its stability is the best when optimizing 13 benchmark test functions. Moreover, compared to other listed algorithms, the proposed algorithm performs best in terms of convergence speed, search accuracy, and robustness. The optimization results for a part of CEC2017 test functions show that the proposed algorithm performs best in the optimization of hybrid and composition functions. It means that the proposed algorithm has great potential in dealing with complex combinatorial problems. Finally, the proposed algorithm is successfully applied to solve six practical constrained engineering problems. The results are better than the listed advanced algorithms. It shows that the proposed algorithm has excellent optimization ability and scalability. However, the proposed algorithm increases the time complexity and the average running time is relatively long. The algorithm still has some room for improvement.

In future work, we will focus on the following tasks:We will prove the convergence and stability of the proposed IWBOA theoretically.We will apply the IBWOA to solve the wind power prediction problem.

## Figures and Tables

**Figure 1 entropy-24-01640-f001:**
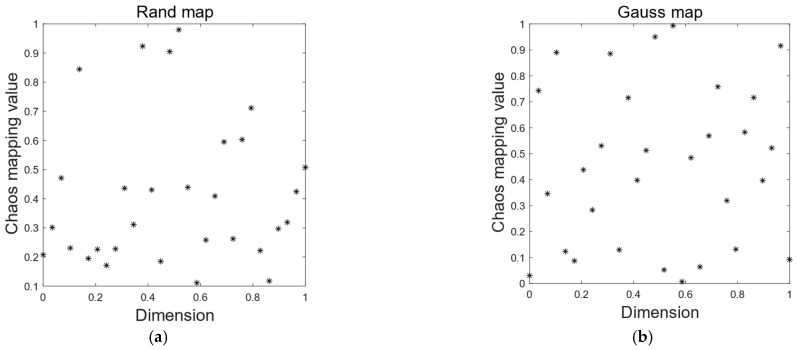
Two different ways of initializing the population. (**a**) Rand initializing the individual distribution. (**b**) Gauss chaotic sequence distribution.

**Figure 2 entropy-24-01640-f002:**
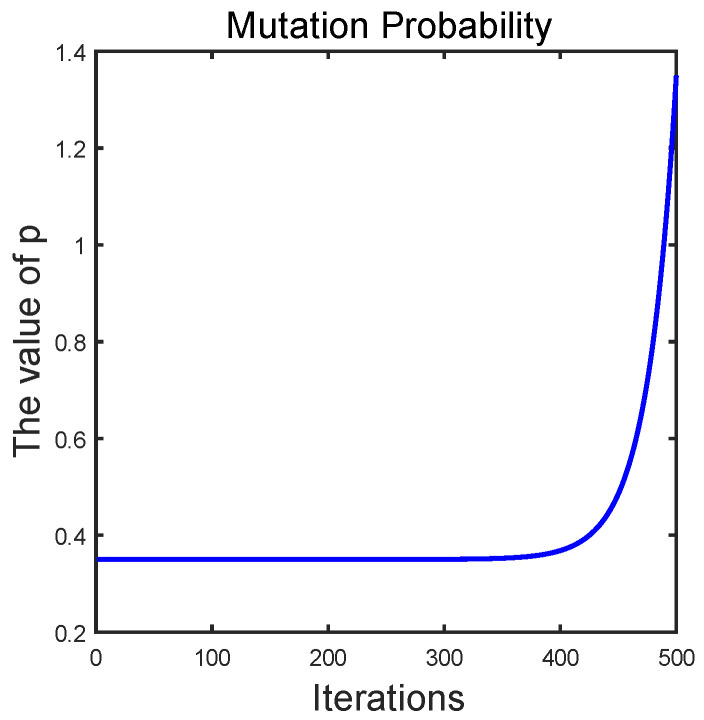
Mutation probability curve.

**Figure 3 entropy-24-01640-f003:**
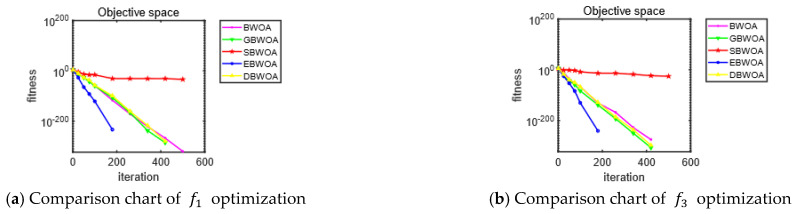
Convergence curves of each improved algorithm for representative test functions in 30 dimensions.

**Figure 4 entropy-24-01640-f004:**
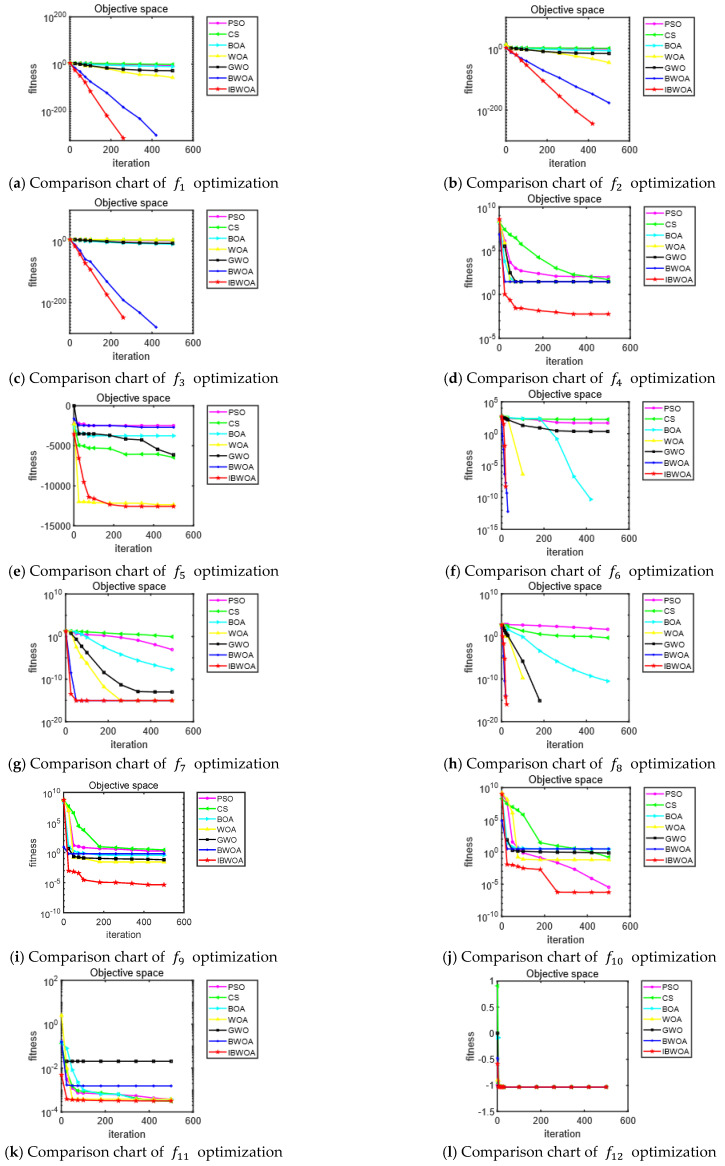
Convergence curves of seven algorithms for 13 test functions.

**Table 1 entropy-24-01640-t001:** Information of benchmark function.

Function Name	Expressions	Range	Optimal	Accept
Sphere Model	$f_{1}\left(x\right)=\sum_{i=1}^{n}x_{i}^{2}$	[−100, 100]	0	1.00 × 10^−3^
Schwefel’s problem 2.22	$f_{2}\left(x\right)=\sum_{i=1}^{n}\left|x_{i}\right|+\prod_{i=1}^{n}\left|x_{i}\right|$	[−10, 10]	0	1.00 × 10^−3^
Schwefel’s problem 1.2	$f_{3}\left(x\right)=\sum_{i=1}^{n}{\left(\sum_{i=1}^{i}x_{j}\right)}^{2}$	[−100, 100]	0	1.00 × 10^−3^
Generalized Rosenbrock’s	$f_{4}\left(x\right)=\sum_{i=1}^{n-1}\left[100{\left(x_{i+1}-x_{i}^{2}\right)}^{2}+{\left(x_{i}-1\right)}^{2}\right]$	[−30, 30]	0	1.00 × 10^−2^
Generalized Schwefel’s problem 2.26	$f_{5}\left(x\right)=\sum_{i=1}^{n}-x_{i}\sin\sqrt{\left|x_{i}\right|}$	[−500, 500]	−418.9829n	1.00 × 10^2^
Generalized Rastrigin’s	$f_{6}\left(x\right)=\sum_{i=1}^{n}\left[x_{i}^{2}-10\cos\left(2\pi x_{i}\right)+10\right]$	[−5.12, 5.12]	0	1.00 × 10^−2^
Ackley’s Function	$f_{7}\left(x\right)=-20\exp\left(-0.2\sqrt{\frac{1}{n}\sum_{i=1}^{n}x_{i}^{2}}\right)-\exp\left(\frac{1}{n}\sum_{i=1}^{n}\cos\left(2\pi x_{i}\right)\right)+20+e$	[−32, 32]	0	1.00 × 10^−2^
Generalized Griewank	$f_{8}\left(x\right)=\frac{1}{4000}\sum_{i=1}^{n}x_{i}^{2}-\prod_{i=1}^{n}\cos\frac{x_{i}}{\sqrt{i}}+1$	[−600, 600]	0	1.00 × 10^−2^
Generalized Penalized	$f_{9}=\frac{\pi }{n}\left\{10\sin\left(\pi y_{1}\right)+\sum_{i=1}^{n-1}{\left(y_{i}-1\right)}^{2}\left[1+10\sin^{2}\left(\pi y_{i+1}\right)\right]+{\left(y_{n}-1\right)}^{2}\right\}+\sum_{i=1}^{n}u\left(x_{i},10,100,4\right)$ $y_{i}=1+\frac{x_{i}+1}{4},u\left(x_{i},a,k,m\right)=\{\begin{array}{c} k{\left(x_{i}-a\right)}^{2},x_{i}>a\\ 0,-a<x_{i}<a\\ k{\left(-x_{i}-a\right)}^{m},x_{i}<-a \end{array}$	[−50, 50]	0	1.00 × 10^−2^
Generalized Penalized 2	$f_{10}\left(x\right)=0.1\left\{\sin^{2}\left(3\pi x_{1}\right)+\sum_{i=1}^{n}{\left(x_{i}-1\right)}^{2}\left[1+\sin^{2}\left(3\pi x_{i}+1\right)\right]+{\left(x_{n}-1\right)}^{2}\left[1+\sin^{2}\left(2\pi x_{n}\right)\right]\right\}+\sum_{i=1}^{n}u\left(x_{i},5,100,4\right)$	[−50, 50]	0	1.00 × 10^−2^
Kowalik’s Function	$f_{11}\left(x\right)=\sum_{i=1}^{11}{\left[a_{i}-\frac{x_{i}\left(b_{i}^{2}+b_{i}x_{2}\right)}{b_{i}^{2}+b_{i}x_{3}+x_{4}}\right]}^{2}$	[−5, 5]	3.07 ×10^−4^	1.00 × 10^−2^
Six-Hump Camel-Back	$f_{12}\left(x\right)=4x_{1}^{2}-2.1x_{1}^{4}+\frac{1}{3}x_{1}^{6}+x_{1}x_{2}-4x_{2}^{2}+4x_{2}^{4}$	[−5, 5]	−1.0316	1.00 × 10^−2^
Branin	$f_{13}\left(x\right)={\left(x_{2}-\frac{5.1}{4\pi^{2}}x^{2}+\frac{5}{\pi }x_{1}-6\right)}^{2}+\left(1-\frac{1}{8\pi }\right)\cos x_{1}+10$	[−5, 5]	0.398	1.00 × 10^−2^

**Table 2 entropy-24-01640-t002:** Performance Comparison of Improved Strategies in Different Dimensions.

Fun	Algorithm	Dim = 30		Dim = 100		Dim = 500	
		Mean	Std	Mean	Std	Mean	Std
$f_{1}$	BWOA	3.60 × 10^−312^	0	2.36 × 10^−302^	0	2.35 × 10^−298^	0
	GBWOA	3.53 × 10^−312^	0	5.46 × 10^−310^	0	3.96 × 10^−299^	0
	SBWOA	1.53 × 10^−14^	8.16 × 10^−14^	1.46 × 10^−14^	6.82 × 10^−14^	1.79 × 10^−14^	9.33 × 10^−14^
	EBWOA	**0**	**0**	**0**	**0**	**0**	**0**
	DBWOA	1.26 × 10^−306^	0	2.73 × 10^−297^	0	9.88 × 10^−307^	0
$f_{3}$	BWOA	5.24 × 10^−320^	0	5.82 × 10^−302^	0	1.25 × 10^−299^	0
	GBWOA	1.60 × 10^−320^	0	5.77 × 10^−310^	0	4.75 × 10^−308^	0
	SBWOA	1.96 × 10^−10^	1.02 × 10^−9^	5.57 × 10^−8^	3.00 × 10^−7^	3.27 × 10^−7^	1.80 × 10^−6^
	EBWOA	**0**	**0**	**0**	**0**	**0**	**0**
	DBWOA	9.78 × 10^−297^	0	1.40 × 10^−297^	0	1.26 × 10^−293^	0
$f_{5}$	BWOA	−4.48 × 10^3^	8.68 × 10^2^	−9.05 × 10^3^	2.24 × 10^3^	−1.89 × 10^4^	3.64 × 10^3^
	GBWOA	−8.09 × 10^3^	1.71 × 10^3^	−2.80 × 10^4^	5.61 × 10^3^	−1.30 × 10^5^	3.57 × 10^4^
	SBWOA	**−** **1.23 × 10^4^**	**7.08 × 10^2^**	**−** **4.08 × 10^4^**	**2.65 × 10^3^**	**−2.07 × 10^5^**	**7.50 × 10^3^**
	EBWOA	−6.89 × 10^3^	7.02 × 10^2^	−1.54 × 10^4^	1.91 × 10^3^	−3.61 × 10^4^	4.50 × 10^3^
	DBWOA	−5.84 × 10^3^	1.17 × 10^3^	−1.12 × 10^4^	2.46 × 10^3^	2.53 × 10^4^	7.65 × 10^3^
$f_{9}$	BWOA	8.41 × 10^−1^	2.56 × 10^−1^	1.11 × 10^0^	1.11 × 10^−1^	1.18 × 10^0^	2.05 × 10^−2^
	GBWOA	2.14 × 10^−1^	2.00 × 10^−1^	2.00 × 10^−1^	1.28 × 10^−1^	1.65 × 10^−1^	2.00 × 10^−2^
	SBWOA	**1.75 × 10^−7^**	**2.48 × 10^−7^**	**3.45 × 10^−7^**	**1.54 × 10^−7^**	**4.61 × 10^−7^**	**3.18 × 10^−6^**
	EBWOA	4.56 × 10^−1^	1.77 × 10^−1^	7.52 × 10^−1^	8.90 × 10^−2^	1.02 × 10^−1^	2.90 × 10^−2^
	DBWOA	8.41 × 10^−1^	2.79 × 10^−1^	1.11 × 10^0^	9.64 × 10^−2^	1.17 × 10^0^	1.78 × 10^−2^

(The bold numbers in a table are the best results, which are shown in each row of the table).

**Table 3 entropy-24-01640-t003:** Comparison of average runtime and success rate of 13 benchmarking functions for optimization.

Fun	BWOA		GBWOA		DBWOA		SBWOA		EBWOA		IBWOA	
	Y	$P_{c}$	Y	$P_{c}$	Y	$P_{c}$	Y	$P_{c}$	Y	$P_{c}$	Y	$P_{c}$
$f_{1}$	5.92 × 10^−2^	100%	6.31 × 10^−2^	100%	5.55 × 10^−2^	100%	5.98 × 10^−2^	100%	4.88 × 10^−1^	100%	3.18 × 10^−1^	100%
$f_{2}$	7.15 × 10^−2^	100%	7.16 × 10^−2^	100%	6.62 × 10^−2^	100%	6.87 × 10^−2^	100%	5.06 × 10^−1^	100%	3.32 × 10^−1^	100%
$f_{3}$	6.19 × 10^−1^	100%	6.11 × 10^−1^	100%	6.27 × 10^−1^	100%	6.00 × 10^−1^	100%	1.98 × 10^0^	100%	1.39 × 10^0^	100%
$f_{4}$	9.13 × 10^−2^	0%	9.07 × 10^−2^	0%	1.16 × 10^−1^	0%	9.16 × 10^−2^	83.3%	5.61 × 10^−1^	0%	3.70 × 10^−1^	90%
$f_{5}$	1.14 × 10^−1^	0%	1.17 × 10^−1^	0%	1.02 × 10^−1^	0%	1.23 × 10^−1^	70%	5.84 × 10^−1^	0%	3.90 × 10^−1^	80%
$f_{6}$	7.75 × 10^−2^	100%	7.47 × 10^−2^	100%	7.72 × 10^−2^	100%	7.73 × 10^−2^	100%	4.78 × 10^−1^	100%	3.26 × 10^−1^	100%
$f_{7}$	1.00 × 10^−1^	100%	9.91 × 10^−2^	100%	1.07 × 10^−1^	100%	8.90 × 10^−2^	100%	5.09 × 10^−1^	100%	3.46 × 10^−1^	100%
$f_{8}$	1.22 × 10^−1^	100%	1.25 × 10^−1^	100%	1.47 × 10^−1^	100%	1.13 × 10^−1^	100%	5.99 × 10^−1^	100%	4.03 × 10^−1^	100%
$f_{9}$	1.50 × 10^−1^	0%	1.41 × 10^−1^	0%	2.08 × 10^−1^	0%	1.45 × 10^−1^	100%	6.92 × 10^−1^	0%	4.65 × 10^−1^	100%
$f_{10}$	1.34 × 10^−1^	0%	1.43 × 10^−1^	0%	1.49 × 10^−1^	0%	1.50 × 10^−1^	100%	6.60 × 10^−1^	0%	4.74 × 10^−1^	100%
$f_{11}$	5.76 × 10^−2^	40%	5.68 × 10^−2^	63.3%	5.65 × 10^−2^	40%	6.12 × 10^−2^	63.3%	2.00 × 10^−1^	53.3%	1.44 × 10^−1^	100%
$f_{12}$	4.81 × 10^−2^	86.6%	4.71 × 10^−2^	86.6%	4.42 × 10^−2^	100%	4.66 × 10^−2^	100%	1.53 × 10^−1^	73.3%	1.09 × 10^−1^	100%
$f_{13}$	5.11 × 10^−2^	100%	4.51 × 10^−2^	100%	4.59 × 10^−2^	100%	4.04 × 10^−2^	100%	1.12 × 10^−1^	100%	9.35 × 10^−2^	100%

**Table 4 entropy-24-01640-t004:** Algorithm Parameter Setting.

Algorithm	Parameter
PSO	$c_{1}=c_{2}=2,w\in \left[0.2,0.9\right],V_{\max}=1,V_{\min}=-1$
GWO	$a_{\mathrm{first}}=2,a_{\mathrm{fianl}}=0$
WOA	b=1
BOA	$a=0.1,p=0.6,c_{0}=0.01$
CS	$P_{a}=0.25$
BWOA	—
IBWOA	—

**Table 5 entropy-24-01640-t005:** Comparison of Optimization Results of 7 Algorithms in 30 Dimensions.

Fun	PSO		CS		BOA		WOA		GWO		BWOA		IBWOA	
	Mean	Std	Mean	Std	Mean	Std	Mean	Std	Mean	Std	Mean	Std	Mean	Std
$f_{1}$	9.03 × 10^−7^	1.35 × 10^−6^	5.02 × 10^−39^	1.65 × 10^−38^	1.41 × 10^−11^	1.25 × 10^−12^	1.41 × 10^−30^	4.91 × 10^−30^	6.05 × 10^−34^	1.14 × 10^−33^	3.60 × 10^−312^	0	**0**	**0**
$f_{2}$	2.02 × 10^−3^	2.58 × 10^−3^	3.77 × 10^−20^	7.77 × 10^−20^	5.58 × 10^−9^	6.32 × 10^−10^	1.06 × 10^−21^	2.39 × 10^−21^	2.37 × 10^−20^	2.37 × 10^−20^	7.19 × 10^−158^	2.69 × 10^−150^	**0**	**0**
$f_{3}$	6.41 × 10^0^	3.40 × 10^0^	5.48 × 10^−38^	2.07 × 10^−37^	1.17 × 10^−11^	1.42 × 10^−12^	5.39 × 10^−7^	2.93 × 10^−6^	1.98 × 10^−7^	7.35 × 10^−7^	5.24 × 10^−320^	0	**0**	**0**
$f_{4}$	9.67 × 10^1^	6.01 × 10^1^	3.95 × 10^1^	2.36 × 10^1^	2.88 × 10^1^	3.13 × 10^−2^	2.79 × 10^1^	7.64 × 10^−1^	2.68 × 10^1^	6.99 × 10^1^	2.90 × 10^1^	2.56 × 10^−2^	**5.46 × 10^−3^**	**8.83 × 10^−3^**
$f_{5}$	−4.84 × 10^3^	1.15 × 10^3^	−1.09 × 10^2^	1.08 × 10^1^	−2.26 × 10^3^	4.56 × 10^2^	−5.01 × 10^3^	7.00 × 10^2^	−6.12 × 10^3^	−4.09 × 10^3^	−4.48 × 10^3^	8.68 × 10^2^	**−1.25 × 10^4^**	**7.94 × 10^1^**
$f_{6}$	5.01 × 10^1^	1.44 × 10^1^	0	0	5.23 × 10^1^	8.55 × 10^1^	0	0	1.39 × 10^0^	3.21 × 10^0^	0	0	**0**	**0**
$f_{7}$	3.29 × 10^−4^	2.45 × 10^−4^	8.88 × 10^−16^	0	5.38 × 10^−9^	1.13 × 10^−9^	7.40 × 10^0^	9.90 × 10^0^	4.27 × 10^−14^	3.81 × 10^−15^	8.88 × 10^−16^	0	**8.88 × 10^−16^**	**0**
$f_{8}$	2.03 × 10^1^	5.88 × 10^0^	0	0	9.02 × 10^−13^	8.90 × 10^−13^	2.90 × 10^−4^	1.59 × 10^−3^	3.54 × 10^−3^	7.24 × 10^−3^	0	0	**0**	**0**
$f_{9}$	6.92 × 10^−3^	2.63 × 10^−2^	3.15 × 10^0^	1.27 × 10^0^	6.05 × 10^−1^	1.57 × 10^−1^	3.40 × 10^−1^	2.15 × 10^−1^	5.34 × 10^−2^	2.07 × 10^−2^	8.41 × 10^−1^	2.56 × 10^−1^	**2.16 × 10^−6^**	**2.56 × 10^−6^**
$f_{10}$	6.68 × 10^−3^	8.91 × 10^−3^	5.71 × 10^0^	2.09 × 10^0^	2.81 × 10^0^	2.05 × 10^−1^	1.89 × 10^0^	2.66 × 10^−1^	6.54 × 10^−1^	4.47 × 10^−3^	2.95 × 10^0^	1.68 × 10^−1^	**3.81 × 10^−5^**	**3.55 × 10^−5^**
$f_{11}$	5.77 × 10^−4^	2.22 × 10^−4^	4.19 × 10^−4^	1.22 × 10^−4^	4.81 × 10^−4^	1.28 × 10^−4^	5.72 × 10^−4^	3.24 × 10^−4^	3.82 × 10^−3^	7.41 × 10^−3^	4.26 × 10^−3^	5.28 × 10^−3^	**3.10 × 10^−4^**	**1.64 × 10^−5^**
$f_{12}$	−1.0316	6.2 × 10^−16^	−1.0316	6.8 × 10^−15^	−1.0316	6.6 × 10^−15^	−1.0316	4.2 × 10^−7^	−1.0316	7.7 × 10^−8^	−1.0272	1.24 × 10^−2^	**−1.0316**	**7.78 × 10^−8^**
$f_{13}$	0.398	0	0.398	4.91 × 10^−12^	0.404	0.015	0.398	2.7 × 10^−5^	0.398	1.61 × 10^−5^	0.553	8.33 × 10^−1^	**0.398**	**0**

(The bold numbers in a table are the best results, which are shown in each row of the table).

**Table 6 entropy-24-01640-t006:** Wilcoxon rank sum detection results.

Fun	PSO	CS	BOA	WOA	GWO	BWOA
$f_{1}$	1.21 × 10^−12^	1.21 × 10^−12^	1.21 × 10^−12^	1.21 × 10^−12^	1.21 × 10^−12^	2.16 × 10^−2^
$f_{2}$	1.21 × 10^−12^	1.21 × 10^−12^	1.21 × 10^−12^	1.21 × 10^−12^	1.21 × 10^−12^	1.21 × 10^−12^
$f_{3}$	1.21 × 10^−12^	1.21 × 10^−12^	1.21 × 10^−12^	1.21 × 10^−12^	1.21 × 10^−12^	2.16 × 10^−2^
$f_{4}$	3.02 × 10^−11^	3.02 × 10^−11^	3.02 × 10^−11^	3.02 × 10^−11^	3.02 × 10^−11^	3.02 × 10^−11^
$f_{5}$	3.02 × 10^−11^	3.02 × 10^−11^	3.02 × 10^−11^	3.02 × 10^−11^	3.02 × 10^−11^	3.02 × 10^−11^
$f_{6}$	1.21 × 10^−12^	NaN	1.70 × 10^−8^	NaN	1.21 × 10^−12^	NaN
$f_{7}$	1.21 × 10^−12^	NaN	1.21 × 10^−12^	1.21 × 10^−12^	1.21 × 10^−12^	NaN
$f_{8}$	1.21 × 10^−12^	NaN	1.21 × 10^−12^	1.21 × 10^−12^	1.21 × 10^−12^	NaN
$f_{9}$	838 × 10^−7^	3.02 × 10^−11^	3.02 × 10^−11^	3.02 × 10^−11^	3.02 × 10^−11^	3.02 × 10^−11^
$f_{10}$	1.41 × 10^−4^	3.02 × 10^−11^	3.02 × 10^−11^	3.02 × 10^−11^	3.02 × 10^−11^	3.02 × 10^−11^
$f_{11}$	1.54 × 10^−1^	8.89 × 10^−10^	6.74 × 10^−6^	3.26 × 10^−7^	3.51 × 10^−2^	9.76 × 10^−10^
$f_{12}$	2.36 × 10^−12^	1.45 × 10^−11^	1.82 × 10^−9^	5.86 × 10^−6^	5.19 × 10^−2^	3.02 × 10^−11^
$f_{13}$	1.21 × 10^−12^	1.21 × 10^−12^	2.37 × 10^−10^	6.05 × 10^−7^	1.03 × 10^−2^	1.37 × 10^−1^
+/=/−	12/0/1	10/3/0	13/0/0	12/1/0	12/0/1	9/3/1

**Table 7 entropy-24-01640-t007:** Information of part CEC2017 function.

Fun	Dim	Type	Range	Optimal
CEC03	10	UN	[−100, 100]	300
CEC04	10	MF	[−100, 100]	400
CEC05	10	MF	[−100, 100]	500
CEC06	10	MF	[−100, 100]	600
CEC08	10	MF	[−100, 100]	800
CEC011	10	HF	[−100, 100]	1100
CEC016	10	HF	[−100, 100]	1600
CEC017	10	HF	[−100, 100]	1700
CEC020	10	HF	[−100, 100]	2000
CEC021	10	CF	[−100, 100]	2100
CEC023	10	CF	[−100, 100]	2300
CEC024	10	CF	[−100, 100]	2400
CEC025	10	CF	[−100, 100]	2500

**Table 8 entropy-24-01640-t008:** Comparison of CEC2017 function optimization results.

Fun		PRO	WOA	GWO	BWOA	IBWOA
CEC03	Max	2.13 × 10^4^	6.10 × 10^3^	3.34 × 10^3^	2.07 × 10^4^	**1.71 × 10^3^**
	Min	6.01 × 10^3^	**3.20 × 10^2^**	3.82 × 10^2^	4.43 × 10^3^	3.33 × 10^2^
	Mena	1.47 × 10^4^	9.05 × 10^2^	1.34 × 10^3^	1.20 × 10^4^	**6.54 × 10^2^**
	Std	4.04 × 10^3^	1.15 × 10^3^	8.80 × 10^2^	4.19 × 10^3^	**2.98 × 10^2^**
	Rank	5	2	3	4	1
CEC04	Max	1.86 × 10^3^	5.79 × 10^2^	**4.63 × 10^2^**	1.27 × 10^3^	5.17 × 10^2^
	Min	4.71 × 10^2^	4.02 × 10^2^	4.03 × 10^2^	4.22 × 10^2^	**4.00 × 10^2^**
	Mena	1.06 × 10^3^	4.43 × 10^2^	4.12 × 10^2^	5.80 × 10^2^	**4.08 × 10^2^**
	Std	4.48 × 10^2^	5.10 × 10^1^	**1.27 × 10^1^**	1.43 × 10^2^	1.59 × 10^1^
	Rank	5	3	2	4	1
CEC05	Max	6.33 × 10^2^	5.94 × 10^2^	**5.23 × 10^2^**	6.27 × 10^2^	6.01 × 10^2^
	Min	5.47 × 10^2^	5.19 × 10^2^	**5.06 × 10^2^**	5.17 × 10^2^	5.21 × 10^2^
	Mena	5.93 × 10^2^	5.54 × 10^2^	**5.12 × 10^2^**	5.57 × 10^2^	5.54 × 10^2^
	Std	2.24 × 10^1^	2.13 × 10^1^	**4.99 × 10^0^**	2.06 × 10^1^	2.37 × 10^1^
	Rank	5	2	1	4	3
CEC06	Max	6.80 × 10^2^	6.52 × 10^2^	**6.01 × 10^2^**	6.72 × 10^2^	6.60 × 10^2^
	Min	6.24 × 10^2^	6.12 × 10^2^	**6.00 × 10^2^**	6.13 × 10^2^	6.02 × 10^2^
	Mena	6.52 × 10^2^	6.31 × 10^2^	**6.00 × 10^2^**	6.43 × 10^2^	6.25 × 10^2^
	Std	1.38 × 10^1^	1.23 × 10^1^	**2.43 × 10^−1^**	1.32 × 10^1^	1.30 × 10^1^
	Rank	5	3	1	4	2
CEC08	Max	8.94 × 10^2^	8.71 × 10^2^	8.32 × 10^2^	8.84 × 10^2^	**8.16 × 10^2^**
	Min	8.29 × 10^2^	8.18 × 10^2^	**8.06 × 10^2^**	8.21 × 10^2^	8.11 × 10^2^
	Mena	8.66 × 10^2^	8.40 × 10^2^	8.14 × 10^2^	8.56 × 10^2^	**8.13 × 10^2^**
	Std	1.36 × 10^1^	1.48 × 10^1^	6.92 × 10^0^	1.29 × 10^1^	**4.75 × 10^0^**
	Rank	5	3	2	4	1
CEC011	Max	8.91 × 10^3^	1.31 × 10^3^	1.24 × 10^3^	1.22 × 10^4^	**1.23 × 10^3^**
	Min	1.15 × 10^3^	1.11 × 10^3^	1.10 × 10^3^	1.20 × 10^3^	**1.10 × 10^3^**
	Mena	1.93 × 10^3^	1.18 × 10^3^	1.13 × 10^3^	2.32 × 10^3^	**1.12 × 10^3^**
	Std	1.63 × 10^3^	4.70 × 10^1^	3.03 × 10^1^	1.84 × 10^3^	**2.20 × 10^1^**
	Rank	4	3	2	5	1
CEC016	Max	2.61 × 10^3^	2.11 × 10^3^	2.01 × 10^3^	2.39 × 10^3^	**2.01 × 10^3^**
	Min	1.76 × 10^3^	1.61 × 10^3^	1.60 × 10^3^	1.73 × 10^3^	**1.60 × 10^3^**
	Mena	2.09 × 10^3^	1.84 × 10^3^	1.68 × 10^3^	2.05 × 10^3^	**1.66 × 10^3^**
	Std	1.68 × 10^2^	1.32 × 10^2^	8.71 × 10^1^	1.70 × 10^2^	**8.60 × 10^1^**
	Rank	5	3	2	4	1
CEC017	Max	2.04 × 10^3^	1.89 × 10^3^	1.81 × 10^3^	2.05 × 10^3^	**1.80 × 10^3^**
	Min	1.76 × 10^3^	1.74 × 10^3^	1.72 × 10^3^	1.74 × 10^3^	**1.72 × 10^3^**
	Mena	1.88 × 10^3^	1.79 × 10^3^	1.75 × 10^3^	1.84 × 10^3^	**1.74 × 10^3^**
	Std	1.03 × 10^2^	4.36 × 10^1^	**1.90 × 10^1^**	6.28 × 10^1^	1.93 × 10^1^
	Rank	5	3	2	4	1
CEC020	Max	2.48 × 10^3^	2.31 × 10^3^	2.16 × 10^3^	2.42 × 10^3^	**2.10 × 10^3^**
	Min	2.06 × 10^3^	2.03 × 10^3^	**2.01 × 10^3^**	2.05 × 10^3^	2.02 × 10^3^
	Mena	2.21 × 10^3^	2.17 × 10^3^	2.05 × 10^3^	2.22 × 10^3^	**2.04 × 10^3^**
	Std	9.32 × 10^1^	6.35 × 10^1^	3.99 × 10^1^	7.77 × 10^1^	**3.61 × 10^1^**
	Rank	4	3	2	5	1
CEC021	Max	2.41 × 10^3^	2.33 × 10^3^	2.41 × 10^3^	2.40 × 10^3^	**2.33 × 10^3^**
	Min	2.22 × 10^3^	2.20 × 10^3^	2.20 × 10^3^	2.23 × 10^3^	**2.20 × 10^3^**
	Mena	2.36 × 10^3^	2.29 × 10^3^	2.32 × 10^3^	2.35 × 10^3^	**2.28 × 10^3^**
	Std	5.00 × 10^1^	4.08 × 10^1^	5.19 × 10^1^	3.86 × 10^1^	3.89 × 10^1^
	Rank	5	2	3	4	1
CEC023	Max	2.76 × 10^3^	2.63 × 10^3^	2.69 × 10^3^	2.76 × 10^3^	**2.63 × 10^3^**
	Min	2.64 × 10^3^	2.61 × 10^3^	2.62 × 10^3^	2.62 × 10^3^	**2.60 × 10^3^**
	Mena	2.69 × 10^3^	2.62 × 10^3^	2.65 × 10^3^	2.67 × 10^3^	**2.61 × 10^3^**
	Std	3.38 × 10^1^	**7.91 × 10^0^**	1.78 × 10^1^	3.44 × 10^1^	**1.22 × 10^1^**
	Rank	5	2	3	4	1
CEC024	Max	2.97 × 10^3^	2.83 × 10^3^	**2.77 × 10^3^**	2.86 × 10^3^	2.79 × 10^3^
	Min	2.77 × 10^3^	2.51 × 10^3^	2.70 × 10^3^	2.61 × 10^3^	**2.50 × 10^3^**
	Mena	2.84 × 10^3^	2.74 × 10^3^	2.74 × 10^3^	2.78 × 10^3^	**2.69 × 10^3^**
	Std	4.25 × 10^1^	1.03 × 10^2^	**1.31 × 10^1^**	4.57 × 10^1^	1.48 × 10^1^
	Rank	5	3	2	4	1
CEC025	Max	4.68 × 10^3^	3.03 × 10^3^	**2.95 × 10^3^**	3.60 × 10^3^	2.98 × 10^3^
	Min	3.03 × 10^3^	2.90 × 10^3^	2.90 × 10^3^	2.94 × 10^3^	**2.90 × 10^3^**
	Mena	3.70 × 10^3^	2.95 × 10^3^	2.94 × 10^3^	3.07 × 10^3^	**2.93 × 10^3^**
	Std	5.15 × 10^2^	2.71 × 10^1^	1.59 × 10^1^	1.22 × 10^2^	**1.26 × 10^1^**
	Rank	5	3	2	4	1

(The bold numbers in a table are the best results, which shown in each row of the table).

**Table 9 entropy-24-01640-t009:** CEPs information introduction.

Item	Problems	Dim	Cons	Iter
CEP1	Welded beam design	4	7	1000
CEP2	Tension spring design	3	4	1000
CEP3	Three-bar truss design	2	3	1000
CEP4	Cantilever beam design	5	1	1000
CEP5	Deflection of I-beam design	4	1	1000
CEP6	Tubular column design	2	6	1000

**Table 10 entropy-24-01640-t010:** Best results of welded beam design.

Algorithm	$x_{1}$	$x_{2}$	$x_{3}$	$x_{4}$	Optimal
RO	0.20368	3.52846	9.00423	0.20724	1.73534
CPSO	0.202369	3.544214	9.048210	0.205723	1.73148
GWO	0.205676	3.478377	9.036810	0.205778	1.726240
WOA	0.205396	3.484293	9.037426	0.206276	1.730499
SSA	0.2057	3.4714	9.0366	0.2057	1.72491
HFBOA	0.205607	3.473369	9.036766	0.205730	1.725080
BWOA	0.183106	3.696771	9.086768	0.206471	1.734269
IBWOA	0.204300	3.273201	9.104938	0.205632	**1.706809**

(The bold numbers in a table are the best results, which are shown in each row of the table).

**Table 11 entropy-24-01640-t011:** Best results of tension/compression springs.

Algorithm	$x_{1}$	$x_{2}$	$x_{3}$	Optimal
PSO	0.015728	0.357644	11.244543	0.0126747
GWO	0.05169	0.356737	11.28885	**0.012666**
WOA	0.051207	0.345215	12.004032	0.0126763
GSA	0.050276	0.323680	13.525410	0.0127022
HFBOA	0.051841	0.360377	11.078153	**0.012666**
BWOA	0.050811	0.335966	12.620450	0.0126818
IBWOA	0.051889	0.361544	11.011088	**0.012666**

(The bold numbers in a table are the best results, which are shown in each row of the table).

**Table 12 entropy-24-01640-t012:** Best results of three-bar truss design.

Algorithm	$x_{1}$	$x_{2}$	Optimal
CS	0.78867	0.40902	263.9716
SSA	0.788665414	0.408275784	263.8958434
HHO	0.788662816	0.408283133	263.8958434
MBA	0.788565	0.4085597	263.8958522
HFBOA	0.78869137	0.408202602	263.895867
BWOA	0.786199557	0.406694123	263.46345931
IBWOA	0.786027200	0.407114772	**263.46343425**

(The bold numbers in a table are the best results, which are shown in each row of the table).

**Table 13 entropy-24-01640-t013:** Best results of cantilever beam design.

Algorithm	$x_{1}$	$x_{2}$	$x_{3}$	$x_{4}$	$x_{5}$	Optimal
CS	6.0089	5.3049	4.5023	3.5077	2.1504	1.33999
SSA	6.015134526	5.309304676	4.495006716	3.5014262863	2.1527879080	1.3399563910
SOS	6.01878	5.30344	4.49587	3.49896	2.15564	1.33996
MMA	6.0100	5.3000	4.4900	3.4900	2.1500	1.3400
HFBOA	6.016838	5.313519	4.495334	3.495149	2.152926	1.339963
BWOA	6.193426	4.793626	4.143571	3.548273	2.397132	1.315144
IBWOA	6.044796	4.805171	4.431811	3.471760	2.196531	**1.307284**

(The bold numbers in a table are the best results, which are shown in each row of the table).

**Table 14 entropy-24-01640-t014:** Best results of I-beam design.

Algorithm	$x_{1}$	$x_{2}$	$x_{3}$	$x_{4}$	Optimal
CS	50	80	0.9	2.321675	0.0130747
SOS	50	80	0.9	2.32179	0.0130741
CSA	49.9999	80	0.9	2.3216715	0.013074119
BWOA	49.9997	79.99787506	1.7591118802	4.9999723672	0.0066278072
IBWOA	49.9996	79.99996414	1.7644811413	4.9999979901	**0.0066260616**

(The bold numbers in a table are the best results, which are shown in each row of the table).

**Table 15 entropy-24-01640-t015:** Best results of tubular column design.

Algorithm	$x_{1}$	$x_{2}$	Optimal
CS	5.45139	0.29196	26.53217
CSA	5.451163397	0.291965509	26.531364472
KH	5.451278	0.291957	26.5314
HFBOA	5.451157	0.291966	26.499503
BWOA	5.462764455	0.291005616	26.518147206
IBWOA	5.4521171299	0.291734575	**26.49633224**

(The bold numbers in a table are the best results, which are shown in each row of the table).

## Data Availability

Not applicable.

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
