# Peer review of "Improved Black Widow Spider Optimization Algorithm Integrating Multiple Strategies"

_entropy, 2022, doi:10.3390/e24111640_

Round 1

Reviewer 1 Report

Non-WOA algorithms' performance should be compared with the proposed algorithm. 

Reviewer 2 Report

The Black Widow Spider Optimization Algorithm is a novel bio-inspired meta-heuristic optimization algorithm that can be used to optimize multiple engineering and scientific problems. The authors propose to improve the quality of the population by combining the gauss chaotic mapping method and the elite opposition-based learning strategy to improve the convergence speed of the algorithm. The experimental results show that the improved algorithm has excellent optimization ability and scalability. For all these reasons I would recommend acceptance of the paper.

Author Response

We have replied via email regarding the copyright of some images.

Thank you for your affirmation of our article. We will keep trying.

Reviewer 3 Report

This manuscript presents multistrategy black widow spider optimization algorithm, in which some operators of other metaheuristic optimization algorithms such as Gauss chaotic mapping, the sine cosine algorithm, the elite opposition-based learning strategy, and differential evolution are implemented.

The introduced method is compared in terms of performance with different benchmark optimization algorithms together with some other engineering problems, and the average run time of the algorithms are analyzed as well. Also, statistical analysis method of Wilcoxon rank sum detection was applied to compare the obtained results. The applied methodology seems to be complete, and the organization of the manuscript is good too. Also, the English writing of the manuscript is good and it can be read easily.

My comments are as follows:

-The literature review in the introduction section uses both past form and present form to analyze previous studies. It is better to use past form of the verbs in this part.

-Since this study is about improved BWOA, it is better to not to mention strategies adopted to improved other algorithms such as particle swarm optimization algorithm, grasshopper optimization algorithm, gray wolf algorithm. So, the explanations regarding these methods should be eliminated from the introduction section.

-There are already published studies reporting enhancement of the BWOA. I suggest that the authors bring these methods in the literature review of the introduction section and analyze their advantages and disadvantages to illustrate a better comparison of the introduced methods with more similar studies. And if practical, present a comparison with these methods in the results and discussion section as well.

- In mentioning the objectives of the study in the introduction section, the (ii)-th highlight should be stating the objective itself rather than addressing the result.

-Most of the conclusion section is similar to the abstract. For example, the first paragraph of the conclusion section exactly repeats the abstract section. It is better to focus on the main findings of the study in the conclusion section.

-There are also some minor writing issues that need to be alleviated. To mention some, the first sentence of the abstract, the phrase “provide higher a higher level” in section 2.2, and the first sentence in section 4.5.
